# Depuration of Asian Green Mussels Using Chitooligosaccharide-Epigallocatechin Gallate Conjugate: Shelf-Life Extension, Microbial Diversity, and Quality Changes during Refrigerated Storage

**DOI:** 10.3390/foods13193104

**Published:** 2024-09-28

**Authors:** Jirayu Buatong, Nooreeta Bahem, Soottawat Benjakul, Umesh Patil, Avtar Singh

**Affiliations:** 1International Center of Excellence in Seafood Science and Innovation (ICE-SSI), Faculty of Agro-Industry, Prince of Songkla University, Hat Yai, Songkhla 90110, Thailand; jirayu.b@psu.ac.th (J.B.); nooreetabahem@gmail.com (N.B.); soottawat.b@psu.ac.th (S.B.); umesh.p@psu.ac.th (U.P.); 2Department of Food and Nutrition, Kyung Hee University, Seoul 02447, Republic of Korea

**Keywords:** Asian green mussel, food poisoning, chitooligosaccharides, conjugations, microbial population

## Abstract

The effect of chitooligosaccharide-EGCG conjugate (CEC) at different concentrations (0, 1, and 2%; *w*/*v*) and depuration times (DT; 3, and 6 h) on the total viable count and *Vibrio* spp. count of Asian green mussels (AGMs) was studied. Depurated samples showed a reduction in both microbial counts as compared to fresh AGMs (without depuration) and AGMs depurated using water (CON). A similar TVC was noticed at both DTs; however, a lower VC was attained at a DT of 3, irrespective of CEC concentrations (*p* < 0.05). AGMs were depurated for 3 h using 1 and 2% CEC (CE1 and CE2, respectively) solutions and stored for 6 days at 4 °C. The CE2 sample showed the lowest microbial counts as compared to fresh AGMs, CON, and CE1 throughout the storage (*p* < 0.05). CE2 extended the shelf-life of AGMs by 4 days, which was also supported by the lower peroxide value (0.48 mg cumene hydroperoxide/kg sample) and TBARS (0.94 mmol MDA eqv/kg sample) when compared with other samples. Moreover, CE2 had a lower total volatile nitrogen base (TVB; 4.72 mg N/100 g) and trimethylamine (TMA; 3.59 mg N/100 g) on day 4. Furthermore, 2% CEC was able to maintain the DHA content; however, a slightly lower EPA was noticed as compared to the CON. Next-generation sequencing suggested that the CON had a larger microbial community, especially *Vibrio* sp., than the CE2. All the treated samples showed similar likeness scores to the cooked CE2 and CON on day 0. However, slightly lower likeness was attained when CE2 was stored for 4 days, but the likeness score was higher than the acceptable limit (5). No difference in cooking loss was noticed between CON and CE2 samples on day 0. Nevertheless, cooking loss was increased on day 4. Thus, depuration using CEC solution enhanced the shelf-life of AGMs by 4 days without having negative impact on consumer acceptability and textural properties.

## 1. Introduction

Edible mollusks are a diverse group of invertebrates with shells, mainly inhabiting saltwater, freshwater, and land. They include classes like Gastropoda (snails), Bivalvia (clams, scallops, oysters), Cephalopoda (octopus, squid), and Polyplacophora (chitons). Most of the species are consumed worldwide either raw or cooked; among all, calms/mussels are one of the favorite species as a food, especially in Southeast Asian nations. Considering their filter-feeding nature, they have been known to accumulate large amounts of microbes, heavy metals, etc. [1]. Hence, they are prone to spreading foodborne diseases, especially food poisoning, which is associated with eating raw or undercooked seafood. Hence, before consumption, calms/mussels are usually kept in fresh water for the removal of the contaminants present in their body and the process is known as “depuration” [2]. Asian green mussel (AGMs; *Perna viridis*) is considered an economically significant seafood, due to their nutritional and high commercial value around the globe [3,4]. However, considering the high moisture content and local warm conditions in Thailand, they spoil within 2 days without proper storage [5]. Therefore, appropriate processing or storage conditions for AGMs are still required. Generally, AGMs are kept alive by maintaining moist conditions in markets via depuration. Utilizing natural antimicrobial compounds during depuration could be an effective alternative for keeping the mussels alive during transportation and in fresh markets for sale. For example, Palamae et al. [6] used acidic electrolyzed water for depuration of blood clams in combination with high-pressure processing treatment, which extended their shelf-life to 9 days at 4 °C. Palamae et al. [7] used the chitooligosaccharide-catechin conjugate to inhibit *Vibrio parahaemolyticus* in shucked AGMs. However, rare information is available on the depuration of AGMs using natural antimicrobial agents, especially chitooligosaccharide (COS) or its derivatives.

Shrimp shell COS has been extensively studied for its antioxidant, antimicrobial activities, and other human health-related bioactivities. COS has shown excellent potential in preserving fish filets, shrimp, and emulsion systems, especially when conjugated with various polyphenols [8,9]. Among all the polyphenols, epigallocatechin gallate (EGCG) possessed excellent bioactivities, which is associated with the presence of a higher number of hydroxyl groups [10,11]. Additionally, EGCG is known for its high water solubility compared to other polyphenols. EGCG has been incorporated or grafted with various biological molecules, including proteins, polysaccharides, and chitosan [11]. Singh et al. [10] observed an increase in the antimicrobial activity of squid pen COS against pathogenic and spoilage bacteria when conjugated with EGCG (COS-EGCG). In addition, COS-EGCG conjugate demonstrated strong antibacterial activity, particularly against *Listeria monocytogenes* [12], which was more likely due to the inhibition of biofilm formation as well as bacterial motility, and significant cell wall damage, leading to protein leakage and DNA binding. Additionally, Mittal et al. [13] also reported the inhibitory activity of COS-EGCG against α-amylase, showing potential as a natural preservative and functional food ingredient. Therefore, its use as a depurated solution not only enhances the shelf-life of the AGMs, but beneficial for human health after the consumption of mussel meats. Although synthetic antimicrobial agents, such as antibiotics and chemical preservatives, have been widely used in food and medical applications to control bacterial growth, concerns over antimicrobial resistance and potential toxic effects have driven a shift toward natural alternatives. Natural bioactive agents, such as COS and plant-derived compounds, offer a sustainable and safer solution. Thus, exploring alternative natural compounds, such as COS-EGCG, is essential for the development of safer preservation techniques.

The objective of the study was to elucidate the effect of depuration at various times (DT) and concentrations of COS-EGCG conjugate (CEC) solution on the shelf-life of Asian green mussels (AGMs) during refrigerated storage for 6 days. Furthermore, changes in the physicochemical and preliminary sensory properties of mussels stored for the optimum storage time were also elucidated after steam-cooking.

## 2. Materials and Methods

### 2.1. Raw Materials and Chemicals

All chemicals were analytical grade and purchased from Sigma-Aldrich (St. Louis, MO, USA), Oxoid (Thermo Fischer Scientific, Waltham, MA, USA), and CHROMagar™ (Paris, France). Chitosan (molecular weight: ~2.1 × 10^3^ kDa and degree of deacetylation: 85%) was purchased from Marine Bio Resources Co., Ltd., Samutsakhon, Thailand.

### 2.2. Preparation of COS-EGCG Conjugate (CEC)

Firstly, the COS was prepared from shrimp shell chitosan using the ascorbic acid (AsA)/H_2_O_2_ redox pair hydrolysis method [14]. In brief, chitosan (1%, *w*/*v*) was dissolved in acetic acid (2%, *v*/*v*) overnight at room temperature in an Erlenmeyer flask. The pH of the solution was then adjusted to 5.0 using 6 M NaOH. Subsequently, a redox-pair solution of AsA and H_2_O_2_ at a molar ratio of 0.05/0.1 was incubated at 40 °C for 15 min to generate hydroxyl radicals. Thereafter, 2 mL of the redox-pair solution was added to 100 mL of the chitosan solution to initiate hydrolysis for 2 h at 60 °C using a shaker water bath (Memmert, D-91126, Schwabach, Germany). After the reaction, the mixture was allowed to cool down in an ice bath, and the pH was adjusted to 7.0 using 6 M NaOH. Thereafter, the undissolved material was removed by centrifugation (Himac CR22N, Tokyo, Japan) at 10,000× *g* for 15 min at 25 °C. The resulting supernatant was dialyzed against distilled water to remove salt using a dialysis membrane with a cutoff of 500 Da. Then, the dialysate was subjected to lyophilization (Scanvac Model Coolsafe 55, Coolsafe, Lynge, Denmark) to obtain COS powder with a degree of polymerization of 2–8 and an average molecular weight of 0.7 kDa.

Thereafter, CEC was prepared via the H_2_O_2_ and AsA redox pair method as per our previous method [8]. Firstly, COS solution (1%, *w*/*v*) was prepared and adjusted to pH 5.0 using 1 M acetic acid. Simultaneously, 1 M H_2_O_2_ (4 mL) containing 0.10 g of AsA was incubated at 40 °C for 15 min to generate hydroxyl radicals. These two solutions were then combined and incubated at room temperature for 1 h with continuous stirring. Following this, EGCG (0.1%, *w*/*v* of COS) was added to the mixture, which was then incubated at room temperature for 24 h in the dark. The resulting mixture was dialyzed to remove any unreacted EGCG, as described previously, and the dialysate was freeze-dried to obtain COS-EGCG conjugate (CEC) powders (conjugation efficiency: 31%).

### 2.3. Effect of Different Depuration Times (DT) and CEC Concentrations on Total Vailabe Count (TVC) and Vibrio sp. Count (VC) of Asian Green Mussels (AGMs)

Thirty kilograms of live AGMs (average weight 45.9 ± 4.6 g) were purchased from the fresh market in Hat Yai, Thailand. Upon arrival at the laboratory, AGM shells were cleaned and drained on a clean mesh for 10 min. After that, five mussels were dipped into 1000 mL of CEC solution (1, and 2%; *w*/*v*) and depurated for 3, and 6 h at room temperature. The mussels depurated using water were used as a control (CON). The AGMs without depuration treatment were termed as “Fresh AGMs”. The mussels were aerated during the whole depuration process and taken at a designated time, and edible meat was determined for TVC and VC using the spread plate technique method.

Briefly, 25 g of raw edible portions from samples were collected aseptically, diluted, and subjected to microbiological analysis. TVC was determined using plate count agar as described in the FDA BAM (Aerobic Plate Count) method [15]. For VC, a 0.1 mL aliquot of the diluted sample was plated on thiosulfate citrate bile salts sucrose (TCBS) agar to enumerate total *Vibrio* spp. Microbial colonies were counted after incubation at 37 °C for 18–24 h [5].

The DT showing the lowest TVC and VC was selected for further study. As no difference in TVC and VC was noticed between AGMs treated with different DTs (3 and 6 h) at all CEC concentrations (1 and 2%). DT of 3 h was used for further study.

### 2.4. Changes in the Quality of Mussels Depurated Using Different Concentrations of CEC at the Selected DT during Refrigerated Storage

Firstly, mussels were depurated for 3 h, as described previously, at various levels of CE conjugate (1, and 2%; *w*/*v*). Mussels without depuration (fresh AGMs) and with depuration using either water (CON) or CEC solutions were packed in a polystyrene foam tray wrapped with cling film. All the trays were stored at 4 °C for 6 days. The depurated AGMs with 1 and 2% CEC solutions were named CE1 and CE2, respectively. Every second day, the edible portion of each treatment was removed manually and used for analyses.

#### 2.4.1. Microbial Analysis

TVC and VC were determined as described previously. Like TVC, psychrophilic bacterial counts (PBC) were determined, except the agar plates were incubated for 10 days at 4 °C. *Pseudomonas* count (PC) was conducted using a *Pseudomonas* agar base supplemented with cephaloridine, fucidin, and cetrimide. Hydrogen sulfide -producing bacteria count (HSPBC) and *Escherichia coli* (EC) count were assessed using triple sugar iron agar and eosin methylene blue agar, respectively. *Clostridium perfringens* (CC) and lactic acid bacteria (LAB) were enumerated on Perfringens agar base and deMan, Rogosa, and Sharpe agar incubated anaerobically at 35 and 37 °C, respectively, for 3 days. PC and HSPBC plates were incubated at 25 °C for 3 days. For EC plates, incubation was performed for 3 days at 37 °C [16].

#### 2.4.2. Chemical Analysis

Thiobarbituric acid reactive substances (TBARS) or peroxide value (PV) and total volatile nitrogen base content (TVB) or trimethylamine content (TMA) were measured for lipid oxidation and spoilage indicators following the methods given by Phetsang et al. [17].

The CE2 samples on day 4 (CE2-4) showed a lower TVC than the acceptable TVC limit (6 log CFU/g sample) and were selected for further analyses and compared with the CE2 and CON from day 0.

### 2.5. Fatty Acids Analysis

Firstly, oil was extracted from the selected samples using the Bligh and Dyer [18] method, which was then subjected to the preparation of fatty acid methyl esters (FAMEs) using 2 M methanolic NaOH and 2 M methanolic HCl for transmethylation [19]. The FAMEs were injected into gas chromatography (GC) (Agilent 7890B, Santa Clara, CA, USA) with a flame ionization detector (FID). FAMEs dissolved in hexane were injected into the CPSil88 column (100 m × 0.25 mm, df = 0.2 μm) with a split ratio of 1/20. GC analysis was carried out using the following conditions: injection temperature of 250 °C, detector (FID) temperature of 270 °C, and column temperature of 200 °C, gas injection pressure of 31.62 psi, and helium as the carrier gas (constant flow 1.0 mL/min). The peaks were identified based on the retention time of known standards, and the results were expressed as mg/g of lipid. The Supelco 37 Component FAME Mix standard (200–400 μg/mL; Sigma Aldrich, St. Louis, MO, USA) was used for calibration. Only peak areas above 1% of the total graph were considered for the calculation of fatty acid composition following the method of [20]. The identification of individual fatty acids was performed by comparing their retention times with known standards, and the results were expressed as mg/g lipids.

### 2.6. Next-Generation Sequencing

For Next-Generation Sequencing (NGS), CON from day 2 showing the higher microbial load was selected in comparison with CE2 from day 4 (CE2-4). The bacterial population was determined using 16S rRNA gene NGS as tailored by Patil et al. [21]. DNA extraction for all samples was performed using the ZymoBIOMICS-96 MagBead DNA Kit (Zymo Research, Irvine, CA, USA) as per the manufacturer’s instructions. The DNA was amplified at the V3–V4 regions of the 16S rRNA gene using the polymerase chain reaction method. For the NGS process, the Illumina MiSeq platform was employed with a v3 reagent kit (600 cycles) and a 10% PhiX spike-in. The taxonomy assignment was conducted using Uclust from QIIME v1.9.1, with the Zymo Research Database as a reference. QIIME v1.9.1 was also used to analyze composition visualization, alpha diversity, and beta diversity.

### 2.7. Effect of Cooking on the Physicochemical Properties of Depurated AGMs

The selected CEC-treated samples from days 0 (CE2) and 4 (CE2-4) were used for further study in comparison to the CON (fresh mussels depurated with water for 3 h). All samples were steam-cooked for 4–5 min followed by cooling in iced water. Cooled samples were drained for 5 min on a mesh and subjected to the following analyses:

#### 2.7.1. Cooking Loss

Samples were weighed before and after cooking, and cooking loss was computed as follows:Cooking loss (%): [A − B)/A] × 100(1)
where A = initial weight before cooking and B = weight after cooking and cooling.

#### 2.7.2. Appearance and Color

The appearance and color parameters (*L**, *a**, and *b**, representing lightness, redness/greenness, and yellowness/blueness, respectively) for each AGM sample were analyzed using a digital camera and Hunter Lab colorimeter (Colorflex, Reston, VA, USA), respectively.

#### 2.7.3. Texture Profile Analysis

Firmness was assessed through a penetration test, toughness was evaluated using a compression test, and shear force was examined at the middle part of the mussel using Warner–Bratzler blade equipment with the help of a TA-XT2 texture analyzer (Stable Micro Systems, Surrey, UK).

### 2.8. Statistical Analysis

The SPSS package (SPSS 20.0 for Windows, SPSS Inc., Chicago, IL, USA) was used for data analysis. Data obtained from an experiment run in replicate (*n* = 3) were subjected to a one-way analysis of variance (ANOVA) and means were compared using Duncan’s multiple range test (DMRT).

## 3. Results and Discussion

### 3.1. Effect of Different DTs and CEC Concentrations on the TVC and VC of AGMs

The effect of different DTs and depuration solutions (CEC) at different levels on the TVC and VC of AGMs is shown in Figure 1. For fresh AGMs, the TVC was 4.44 log CFU/g samples. After the depuration for 3 h, CE2 had the lowest TVC as compared to CE1 and CON (*p* < 0.05). At a DT of 6 h, an increase in the TVC (5.44 log CFU/g sample) was noticed for the CON sample, which was exceeded by the fresh AGMs, suggesting a faster rate of bacterial growth (*p* < 0.05). On the other hand, CEC-treated mussels (4.02–4.15 log CFU/g sample) showed no change in the count as compared with a DT of 3 h (*p* > 0.05). There was an insignificantly lower TVC in CEC-treated mussels when the DT was increased to 6 h from 3 h. The result suggested that depuration with the CEC solution effectively inhibited the microbial growth. For VC, at a DT of 3 h, CON showed a slightly higher count than the CEC-treated samples (*p* < 0.05). Nevertheless, the VC was similar to that of fresh AGMs (*p* > 0.05). VCs were increased further at 6 h of DT, where no difference was noticed among the CE1 and CON samples (*p* > 0.05). However, CE2 showed the lowest VC at a DT of 6 h (*p* < 0.05) as compared to the remaining samples. In general, mussels are known to carry large microbial community loads due to their filter-feeding habits [1]. Thus, depuration in water is generally performed for all the edible mollusks to remove all heavy metals, microbes, or other contaminants. The decreasing TVC and VC counts were more likely associated with the antimicrobial activity of CEC [10], which could inhibit the growth of bacteria present in the mussel body during the extended DT. There was no change in TVC noticed as the DT increased, regardless of CEC concentration. Furthermore, lower VCs were obtained at a DT of 3 h. Thus, a DT of 3 h was selected for further study.

### 3.2. Changes in Mussel Quality during Refrigerated Storage after CEC Depuration at a Selected DT

#### 3.2.1. Microbial Counts

Changes in the TVC and PBC of AGM meat, as influenced by different concentrations of depuration solution (CEC) at the selected DT (3 h) during refrigerated storage for 6 days, are shown in Figure 2A,B. On day 0, CE2 had the lowest TVC/PBC than the remaining samples (*p* < 0.05), followed by CE1, CON, and fresh AGM samples (*p* > 0.05). Regardless of the samples, TVC/PBC was increased with increasing storage time (*p* < 0.05). However, a sharp increase in TVC/PBC was noticed in the fresh AGM and CON samples on day 2 (*p* < 0.05). The TVC of fresh AGM and CON samples exceeded the acceptable limit of 6 log CFU/g samples on day 2 compared to the treated samples (*p* < 0.05) [22]. A similar result was noticed for PBC, except for CON, which was still lower than the acceptable limit on day 2 (*p* < 0.05). This could be related to the elimination of psychrophilic bacteria during the depuration with water (CON) compared with the non-depurated AGMs (fresh AGMs). Till day 4, CE2 had the lowest TVC/PBC; nevertheless, with further increasing storage day to 6, both CE1 and CE2 surpassed the acceptable limit. On day 6, no difference in TVC was noticed between CE1 and CE2 samples (*p* > 0.05). On the other hand, CE2 had a lower PBC than the CE1 sample on day 6 (*p* < 0.05). The lower microbial count was more likely associated with the antimicrobial activity of CEC, in which microbial destruction was achieved via altering the membrane integrity and alternation in DNA or RAN structure [8,9]. Buatong et al. [12] reported that CEC significantly inhibited the production of extracellular polysaccharides, which played a crucial role in reducing the initial biofilm formation of *Listeria monocytogenes.* This reduction in biofilm formation suggests that the conjugate not only impedes bacterial growth but also interferes with the biofilm structure, which is a key factor in the persistence and resistance of bacterial colonies. Similarly, they also reported DNA damage and protein leakage from the bacterial cells after CEC treatment. In general, hydroxyl and amino groups present in the glucosamine unit of COS are responsible for the bioactivities of COS. Chelation of essential nutrients by EGCG or COS and alteration in the bacterial cell wall integrity via electrostatic interactions between the positively charged amino group (NH_3_^+^) of COS with the negatively charged bacterial membrane are the possible reasons for the bacterial growth inhibition [23,24]. Furthermore, depuration might have resulted in the leaching of microbes from the mussel meat, which could limit the further exponential growth of the bacteria. The result was supported by the lower TVC/PBC in fresh AGMs (non-depurated) sample than in the CON. Overall, CEC solution might penetrate in the mussel body during depuration and provide resistance against bacteria.

For *Vibrio* count (VC), a similar trend was noticed among the different treatments, where CE2 had the lowest VC followed by CE and CON during storage (Figure 3A). The results reconfirmed the antibacterial activity of CEC. However, a decreasing VC was noticed with increasing storage time (*p* < 0.05). This could be due to the depletion of nutrients with increasing spoilage bacteria, as indicated by augmenting TVC/PBC. In addition, *Vibrio* growth is enhanced in the presence of saline water, and the unavailability of saline water could be another reason for the decreasing VC. In general, rising ocean temperatures can increase *Vibrio* spp. in aquatic habitats, leading to bivalve contamination [25,26], which is the most common cause of foodborne diseases. Common pathogens in AGMs include *V. parahaemolyticus*, *V. vulnificus*, and *Shewanella* spp. [27].

For *Pseudomonas* count (PC), CE2 had the lowest PC on day 0, followed by CE1 and CON (*p* < 0.05) (Figure 3B). The CON showed a sharp increase in PC on day 2. However, when storage time was increased to 4 days, no difference in PC was noticed among the CEC-treated samples (*p* > 0.05). This suggested the excellent antimicrobial activity of CEC. In general, *Pseudomonas* has been known as the major bacteria in seafood spoilage [28]. In our previous studies, COS showed excellent potential to damage the *Pseudomonas* bacterial cell wall [29] as well as enhance the shelf-life of Asian seabass filets during refrigerated storage [30]. This could explain the antimicrobial activity of COS, which was enhanced during conjugation with the EGCG [10]. Rao et al. [31] also confirmed the antibacterial activity of COS produced from chitosan by irradiation, which, in combination with lysozyme, inhibited the growth of *E. coli* and *P. fluorescens*. Similarly, COS–lysozyme also enhanced the shelf-life of minced meat for 15 days of refrigerated storage.

Hydrogen sulfide-producing bacterial count (HSPBC) was also increased with increasing storage time, regardless of the CEC treatment. Among all the samples, the CE2 sample had the lowest value, which was also supported by the lower remaining bacterial counts. In general, HSPB are known to be primary contributors to fish spoilage, which produce hydrogen sulfide (H_2_S) as a metabolic byproduct. This contributes to the unpleasant odors and off-flavors, commonly associated with spoiled seafood [32,33].

The *Clostridium perfringens* count (CC), lactic acid bacterial count (LAB), and *Escherichia coli* (EC) count were not detected in all the samples. This suggested that CEC showed excellent potential to inhibit most of the spoilage and pathogenic bacterial strains. However, further application of hurdle technology could be more beneficial to enhance the shelf-life of the mussels. In addition, future studies could be elucidated to determine the penetration of CEC inside the mussel’s body during the depuration, which could be performed via tagging COS or EGCG or their conjugation with the specific fluorescence or dye compounds.

Considering the microbial results, fresh AGMs and CON showed no drastic changes on day 0, hence fresh AGMs were not subjected to further analyses.

#### 3.2.2. Lipid Oxidation

The PV and TBARS values are mainly used to elucidate the production of primary and secondary oxidation products, respectively. On day 0, CE2 had the minimum PV as compared to the CON and CE1 (*p* < 0.05), where no difference in PV was attained among them (*p* > 0.05) (Figure 4A). The result suggested the onset of lipid oxidation during the depuration process. The lower PV could be related to the antioxidant activity of CEC, especially at higher concentrations. The antioxidant activity of CEC is mainly due to the presence of protonated amino groups at C2 and hydroxyl groups at C3 and C6 of the glucosamine unit COS. In addition, hydroxyl groups from EGCG, combined, can act as a hydrogen donor to free radicals [8,10]. Although the direct use of CEC on foods is rarely reported, COS applications as antioxidant agents in Asian seabass slices, Pacific white shrimp, shrimp oil-based emulsions, bivalves, etc. during the extended storage time have been well documented [9,34,35,36]. With further increasing storage time, PV decreased until day 2 for the CE1 and CE2 samples, which suggests the antioxidant activity of penetrated CEC in the mussel body. However, an increase in PV was noticed for the CON sample on the same day (*p* < 0.05). With increasing storage time, on day 4 for all samples, PV increased, and then no change was noticed on day 6 (*p* < 0.05).

TBARS also showed a similar trend as with the PV. However, lower TBARS for CON on day 2 as compared to day 0 could be related to a higher amount of primary oxidation products, as shown in Figure 4B. However, for CEC-treated samples, the decreasing TBARS trend continues until day 4, which could be related to both higher amounts of primary oxidation products and the antioxidant activity of CEC. On day 6, a sharp increase in the TBARS was noticed, suggesting the degradation of primary oxidation products to the secondary ones. The lipid oxidation can be enhanced with the increase in the microbial load, which could produce lipases leading to the generation of free fatty acids. Those free fatty acids are known as excellent prooxidants in various foods [37,38]. Thus, lower lipid oxidation in CEC-treated mussels could also be related to the inhibition of bacterial growth in the samples (Figure 2 and Figure 3). Similar results were noticed in Asian seabass slices when treated with COS [30], where the radicals generated by cold atmospheric plasma were inhibited by the COS at various concentrations. Nanoliposomes loaded with chitosan–EGCG conjugate were also able to limit the lipid oxidation of Asian seabass slices during refrigerated storage [39]. Khan et al. [40] noticed reduction in lipid oxidation of blue mussel (*Mytilus edulis*) by 5 days when treated with ascorbic acid (0.01 M) during ice storage, which is also known as an excellent antioxidant agent.

Thus, CEC was able to retard the lipid oxidation during the depuration as well as during extended storage at refrigerated temperature, especially at higher concentrations.

#### 3.2.3. Freshness of AGMs

TVB and TMA are the freshness indicators for seafood products, especially fish meat. The TVB and TMA were increased with increasing storage time regardless of the types of samples (*p* < 0.05), as shown in Figure 5A and Figure 5B, respectively. On day 0, CON had slightly higher TVB and TMA values than the CEC-treated samples (*p* < 0.05). Among the CEC-treated samples, CE2 had the lowest TVB/TMA values throughout the storage (*p* < 0.05). In general, de-amination of non-protein nitrogenous compounds via the bacterial population resulted in enhanced TVB. TMA, a component of TVB, is produced primarily by the bacterial reduction in trimethylamine oxide [28,30]. These compounds lead to the production of off-odors and flavors that are characteristic of spoilage [17]. The lower TVB/TMA values are supported by the lower microbial load in the CEC-treated samples (Figure 2 and Figure 3). The relationship between microbial load and the formation of TVB and TMA is well documented. Bacteria, particularly those that are psychrotrophic, thrive at refrigeration temperatures and are primarily responsible for the spoilage of fish [41]. Currently, all TVB below the acceptable content for fish are 35 mg N/100 g [42]. Masniyom and Benjama [43] studied the impact of lactic, citric, and acetic acids on the quality of green mussels. Among those acids, lactic acid showed lower levels of TVB and TMA as compared to those dipped in acetic and citric acids, which was linked to microbial growth. Overall, CEC was able to maintain the freshness of the AGMs during refrigeration for 4 days.

The CE2 sample was able to be stored for 4 days without exceeding the acceptable limits of TVC. Thus, the CE2 sample stored for 0 and 4 days was selected for further study in comparison to CON on day 0.

#### 3.2.4. Fatty Acid Content

The fatty acid content of CON and CE2 on day 0 and CE2 on day 4 is shown in Table 1. The chromatograms based on the retention time have also been provided in the Appendix A (Appendix A). Palmitic acid was the most dominant in CE2 sample, followed by the CON sample on day 0, which was decreased during the 4-day storage for the CE2 sample (CE2-4) (*p* < 0.05). A similar result was noticed for other fatty acids present in higher amounts, such as stearic acid, followed by eicosatetraenoic acid and palmitoleic acid. The higher amounts of fatty acids in the CE2 samples as compared to the CON on day 0 were more likely due to the antioxidant and antibacterial activity of CEC, which prevents the lipid oxidation and microbial population that can produce enzymes such as lipase [44]. Fatty acids are particularly susceptible to lipid oxidation and hydrolysis by lipase (produced by bacteria) resulting in the production of free fatty acids, which contribute to off-flavors, such as fishiness or rancidity, in seafood products [45]. The reduction in the fatty acid content on day 4 was more likely related to lipid oxidation and increasing bacterial population, as supported by increasing PV/TBARS and TVC values (Figure 4 and Figure 2, respectively). The result was in agreement with the reduction in monounsaturated fatty acids (MUFA) and polyunsaturated fatty acids (PUFA) after the storage of AGMs for 4 days. However, on day 0, CE2 showed a higher PUFA than the CON sample. Nonetheless, eicosapentaenoic acid (EPA) was present in a slightly lower amount in the CE2 sample than in the CON sample, whereas no difference in docosahexaenoic acid (DHA) was noticed between both samples (*p* > 0.05). Similarly, Singh et al. [29] showed that COS was able to reduce the degradation of the fatty acid composition of Asian seabass fileted during extended storage. Thus, the enhanced preservation of PUFAs, in particular, is significant given their health benefits, including anti-inflammatory properties and cardiovascular protection. This study underscores the potential of CEC treatment not only to maintain fatty acid profiles during storage but also to contribute to the overall quality and safety of seafood products.

### 3.3. Bacterial Diversity

Bacterial diversity of CON and CE2 from days 2 and 4, respectively, was identified by NGS based on the 16S rRNA gene at the family, genera, and species levels as shown in Figure 6, Figure 7 and Figure 8, respectively. Overall, CON showed the highest percentage of bacteria from different family, genera, and species based on the operational taxonomic units (OTUs). Marinilabiaceae (7.24%), *Vibrio* (6.18%), and *Clostridium chauvoei* (5.50%) were the dominant family, genera, and species, respectively. Similarly, for CE2-4 Marinilabiaceae (19.20%), *Ilyobacter-Propionigenium* (4.77%) and *Ilyobacter-Propionigenium insuetus-maris* (4.77%) were the dominant family, genera, and species, respectively. Dominant pathogenic *Vibrio* sp. and *Clostridium chauvoei* were found at higher percentages in CON (2.31 and 5.50%, respectively) than the CE2-4, which were in the range of 0.97 and 4.15%, respectively. In addition, the CON sample was also contaminated with *V. alginolyticus* (0.97%), which was absent in the CE2-4. The result was also supported by higher amounts of family and genera of the respective species in CON compared to the CE2-4 samples. Overall, the CON sample showed a higher percentage of microbial community than the CE2-4, which was supported by the lower bacterial counts (Figure 2 and Figure 3). This could be due to the antimicrobial activity of the CEC, which inhibited microbial growth during refrigerated storage. NGS is a powerful detection technology that allows for efficient and comprehensive identification of microbial communities. However, it is important to note that NGS does not differentiate between living and dead bacteria, which can influence the interpretation of results in terms of active microbial load. Nevertheless, it can be used for better food control and assuring the safety of seafood.

### 3.4. Physicochemical and Textural Properties of Cooked AGMs

#### 3.4.1. Color and Appearance

The color and appearance of cooked samples are shown in Figure 9 (A and B–D, respectively). The highest lightness (*L**) was noticed for the CE2 sample, followed by CON and CE2-4 (*p* < 0.05). A similar trend was noticed for the redness (*a**) and yellowness (*b**) of the samples. The result was supported by the macroscopic images of all three samples (Figure 9B–D). The higher color characteristics suggested the freshness of both CON and CE2 on day 0 compared with the CE2 on day 4. In general, mussels are filter feeders; hence, the color of their tissue mainly depends on the phytoplankton present in the harvested area. Noor et al. [46] reported that the color of female tissues tends to be reddish orange, whereas meat from males is creamy towards orange chocolate. The images obtained were taken from the female mussel as shown in Figure 9B–D. The higher color of CE2 on day 0 was likely associated with the antioxidant activity of the CEC, which might inhibit the oxidation of pigments or carotenoids present in the meat during cooking. Palamae et al. [5] also observed the fading of mussels during different types of cooking. When the storage time was increased to 4 days, the resulting color reduction was associated with the oxidation of pigments as well as their loss via exudation during the extended storage. Thus, CEC was able to maintain the color of the AGM meat during refrigerated storage.

#### 3.4.2. Cooking Loss

Loss of water as well as other nutrients from the foods during cooking is the most common and concerning phenomenon [47]. Weight loss during cooking is shown in Table 2. CON and CE2 samples from day 0 had no noticeable difference in weight loss (*p* > 0.05). However, when CE2 was stored for 4 days, slightly lower weight loss was obtained (*p* < 0.05). This could be associated with the reduction in the water holding capacity of the mussel meat during extended storage, which is mainly related to myofibrillar proteins [48]. However, an increasing microbial load as well as various oxidants could hydrolyze or oxidize those proteins leading to a reduction in the water holding capacity [49]. In addition, different cooking methods can also affect the protein structure, which directly influences the loss of water from mussel meat [48]. Overall, CEC depuration of mussels did not influence the mussels’ meat structure.

#### 3.4.3. Texture Profile Analyses (TPAs)

The TPAs of different AGM samples are shown in Table 2. On day 0, no changes were noticed in the shear force, toughness, and firmness of the CON and CE2 samples (*p* > 0.05). However, increasing storage time to 4 days, a reduction in all the TPA parameters was observed (*p* < 0.05). The shear force, firmness, and toughness of the samples are influenced by myofibrillar as well as collagenous proteins. In addition, muscle fibers and chemical composition, as well as cooking methods, play important roles in the determination of meat/flesh [50,51]. The loss of water, protein, or lipid oxidation could decrease the tenderness of the meat. The lower TPA parameters are also supported by the cooking loss as well as the increasing microbial community, which can hydrolyze the muscle proteins (Figure 2 and Figure 3, and Table 2). Russo et al. [52] also noticed that change in the protein structure and water loss were the main reasons for the reduction in the textural properties of Mediterranean mussels (*Mytilus galloprovincialis*) during different processing and storage conditions. The results suggested that CEC depuration did not result in the changes in the textural properties of AGMs on day 0. However, further investigation could be performed to maintain the textural properties during extended storage times.

#### 3.4.4. Sensorial Properties

Before the commercial application of CEC depuration of mussels, preliminary sensory attributes, such as those shown in Table 2, were determined. Overall, no difference in appearance, color, texture, taste, smell, or overall likeness score was noticed between all the samples (*p* > 0.05) except for the appearance and color attributes of CE2-4, which were lower than the other samples (*p* < 0.05). This could be associated with the oxidation of pigments present in the AGM meat, which might reduce the attractive red-orange color of the samples. The result was also supported by the appearance and color of the CE2-4 sample (Figure 9). In addition, lipid oxidation resulted in the generation of free radicals, which are known to change the color of various meats [53]. Despite these changes, the CEC treatment did not negatively impact the overall eating quality of the AGMs, indicating that it could be a viable method for depuration without compromising sensory attributes.

## 4. Conclusions

Among the different DTs, 3 h inhibited the growth of the TVC and VC of AGMs. When AGMs depurated for 3 h using different concentrations (1 and 2%) of CEC were stored at 4 °C, lower microbial counts were noticed, especially at a higher CEC (2%; CE2) level. Similarly, lower lipid oxidation and TVB/TMA values also supported the extended shelf-life of AGMs of 4 days compared with the control (depurated with water; 2 days). When CE2 and CON were steam-cooked, no changes in cooking loss and textural properties were noticed among them on day 0. On the other hand, during the 4 days of storage, CE2 showed cooking loss and textural and color changes compared with day 0. CEC showed a similar likeness score to that of the CON sample on day 0. Conversely, a slightly lower score was attained for the CE2 sample when stored for 4 days. Nonetheless, the likeness score was higher than the acceptable limit. Thus, depuration of AGMs using CEC, especially at higher concentrations, enhanced the shelf-life without affecting the textural and sensorial properties. However, further application of thermal or non-thermal processing should be studied further. Additionally, studying the effects of CEC treatment over longer storage periods and the penetration of CEC into the mussel body could provide deeper insights into its potential as a universal preservative for various types of seafood.

## 5. Declaration of Generative AI in Scientific Writing

The authors confirm that no AI was used in the direct writing of the text; however, certain sentences were modified to reduce similarity using AI.

## Figures and Tables

**Figure 1 foods-13-03104-f001:**
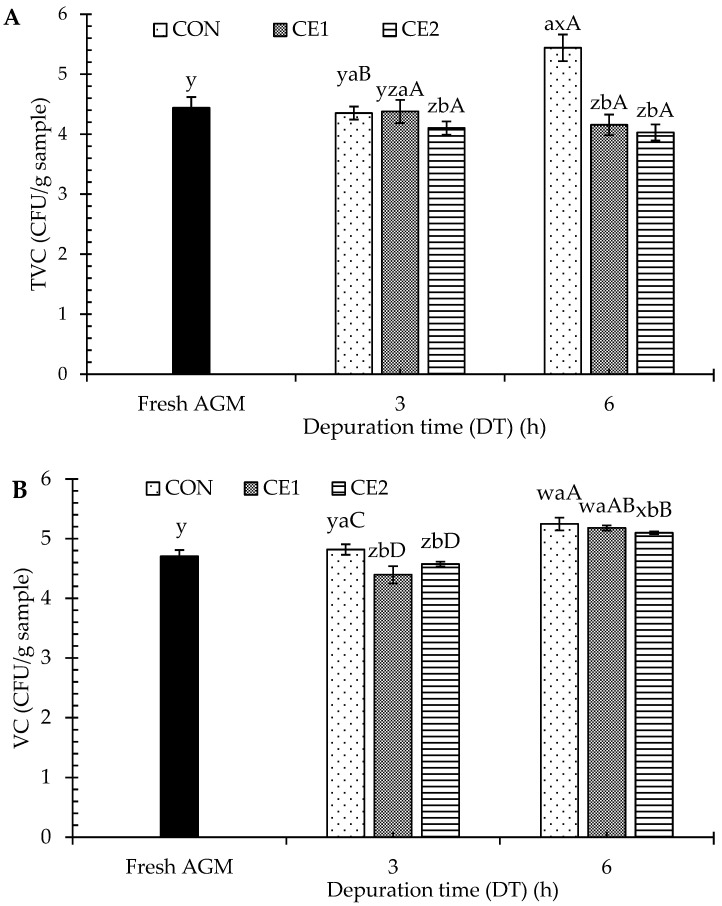
Total viable count (TVC) (**A**) and *Vibrio* sp. count (VC) (**B**) of Asian green mussels subjected to chitooligosaccharide-epigallocatechin gallate conjugate depuration at various concentrations and depuration times. Bars represent the standard deviation (*n* = 3). Different lowercase letters (a and b) on the bars of different samples within the same depuration time (DT) indicate a significant difference (*p* < 0.05). Different lowercase letters (w, x, y, and z) on the bars indicate a significant difference (*p* < 0.05). Different uppercase letters (A, B, C and D) on the bars of same sample from different DTs indicate a significant difference (*p* < 0.05). Fresh AGMs: mussels without depuration; CON: mussels depurated using water; CE1 and CE2: mussels depurated using 1 and 2% chitooligosaccharide-epigallocatechin gallate conjugate (CEC) solution, respectively.

**Figure 2 foods-13-03104-f002:**
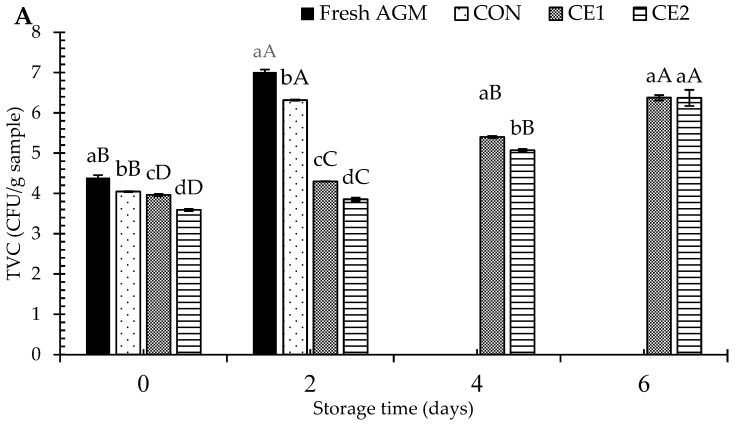
Total viable (**A**) and psychrophilic bacterial counts (**B)** of Asian green mussels (AGMs) subjected to treatment without and with chitooligosaccharide-epigallocatechin gallate conjugate (CEC) depuration at various concentrations. Bars represent the standard deviation (*n* = 3). Different lowercase letters on the bars of different samples within the same storage time indicate a significant difference (*p* < 0.05). Different uppercase letters on the bars of samples from different storage time indicate a significant difference (*p* < 0.05). Fresh AGMs: mussels without depuration; CON: mussels depurated using water; CE1 and CE2: mussels depurated using 1 and 2% chitooligosaccharide-epigallocatechin gallate conjugate (CEC) solution, respectively. TVC and PBC: total viable and psychrophilic bacterial counts, respectively. TVC for the fresh AGM sample was exceeded on day 2; therefore, it was not subjected to further storage. DT was 3 h.

**Figure 3 foods-13-03104-f003:**
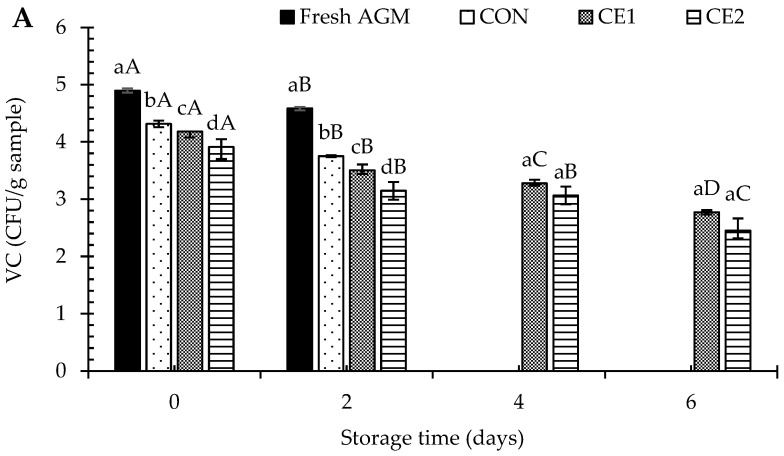
*Vibrio* sp. (VC; (**A**)), *Pseudomonas* (PC; (**B**)), and hydrogen sulfide-producing bacterial (HSPBC; (**C**)) counts of Asian green mussels (AGMs) treated without and with chitooligosaccharide-epigallocatechin gallate conjugate (CEC) depuration at various concentrations. Bars represent the standard deviation (*n* = 3). Different lowercase letters on the bars of different samples within the same storage time indicate a significant difference (*p* < 0.05). Different uppercase letters on the bars of samples from different storage time indicate a significant difference (*p* < 0.05). Fresh AGMs: mussels without depuration; CON: mussels depurated using water; CE1 and CE2: mussels depurated using 1 and 2% chitooligosaccharide-epigallocatechin gallate conjugate (CEC) solution, respectively. PC, EC, HSPBC, CC, and LAB: *Pseudomonas*, *Escherichia coli*, Hydrogen sulfide -producing bacteria, *Clostridium perfringens*, and lactic acid bacteria counts, respectively. TVC for the fresh AGM sample was exceeded on day 2; therefore, it was not subjected to further storage. DT was 3 h.

**Figure 4 foods-13-03104-f004:**
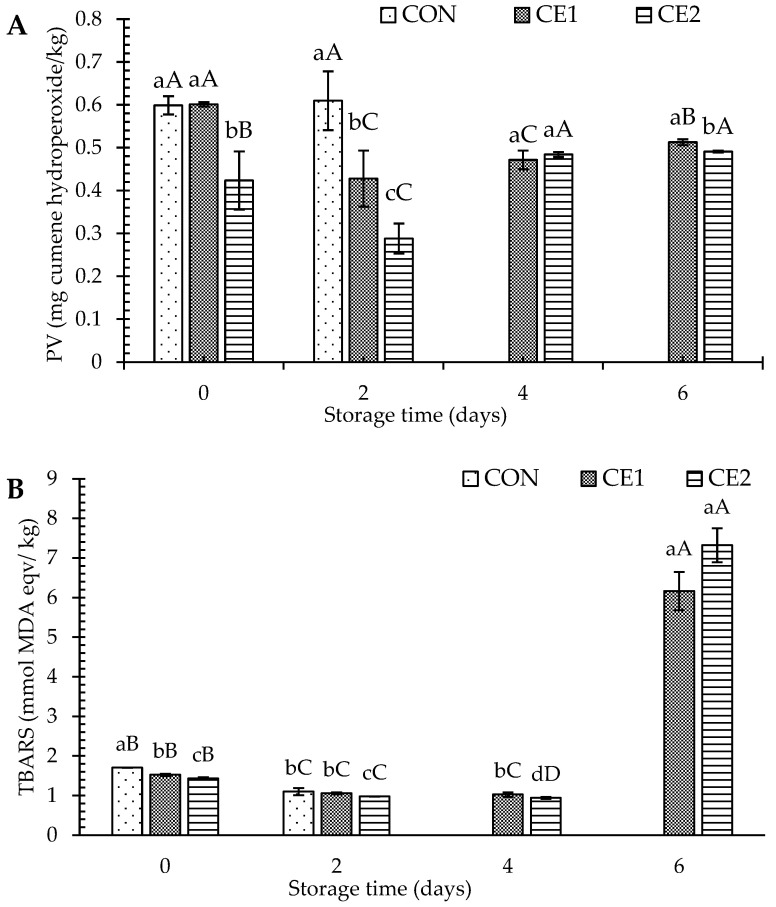
Peroxide value (PV) (**A**) and thiobarbituric acid reactive substances (TBARS) (**B**) of Asian green mussels (AGMs) subjected to chitooligosaccharide-epigallocatechin gallate conjugate (CEC) depuration at various concentrations. Bars represent the standard deviation (*n* = 3). Different lowercase letters on the bars of different samples within the same storage time indicate a significant difference (*p* < 0.05). Different uppercase letters on the bars of samples from different storage time indicate a significant difference (*p* < 0.05). CON: mussels depurated using water; CE1 and CE2: mussels depurated using 1 and 2% chitooligosaccharide-epigallocatechin gallate conjugate (CEC) solution, respectively. PV: peroxide value; TBARS: thiobarbituric acid reactive substances. TVC for the fresh AGM sample was exceeded on day 2; therefore, they were not subjected to further storage. DT was 3 h.

**Figure 5 foods-13-03104-f005:**
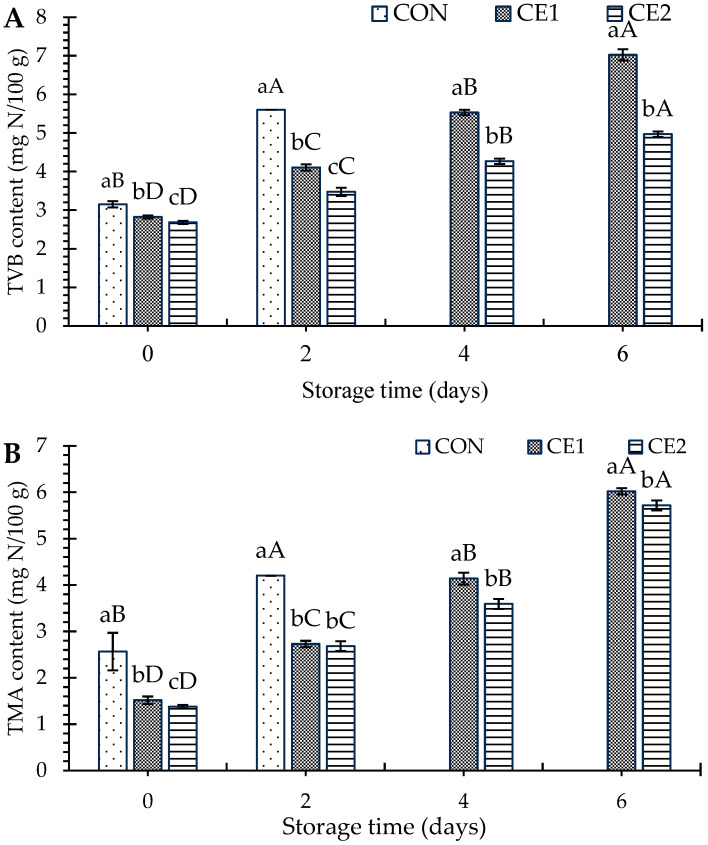
Total volatile nitrogen base content (**A**), and trimethylamine content (**B**) of Asian green mussels (AGMs) subjected to chitooligosaccharide-epigallocatechin gallate conjugate (CEC) depuration at various concentrations for 3 h of depuration. Bars represent the standard deviation (*n* = 3). Different lowercase letters on the bars of different samples within the same storage time indicate a significant difference (*p* < 0.05). Different uppercase letters on the bars of samples from different storage time indicate a significant difference (*p* < 0.05). CON: mussels depurated using water; CE1 and CE2: mussels depurated using 1 and 2% chitooligosaccharide-epigallocatechin gallate conjugate (CEC) solution, respectively. TVB: total volatile bases; TMA: trimethylamine. TVC for the fresh AGM sample was exceeded on day 2; therefore, they were not subjected to further storage. DT was 3 h.

**Figure 6 foods-13-03104-f006:**
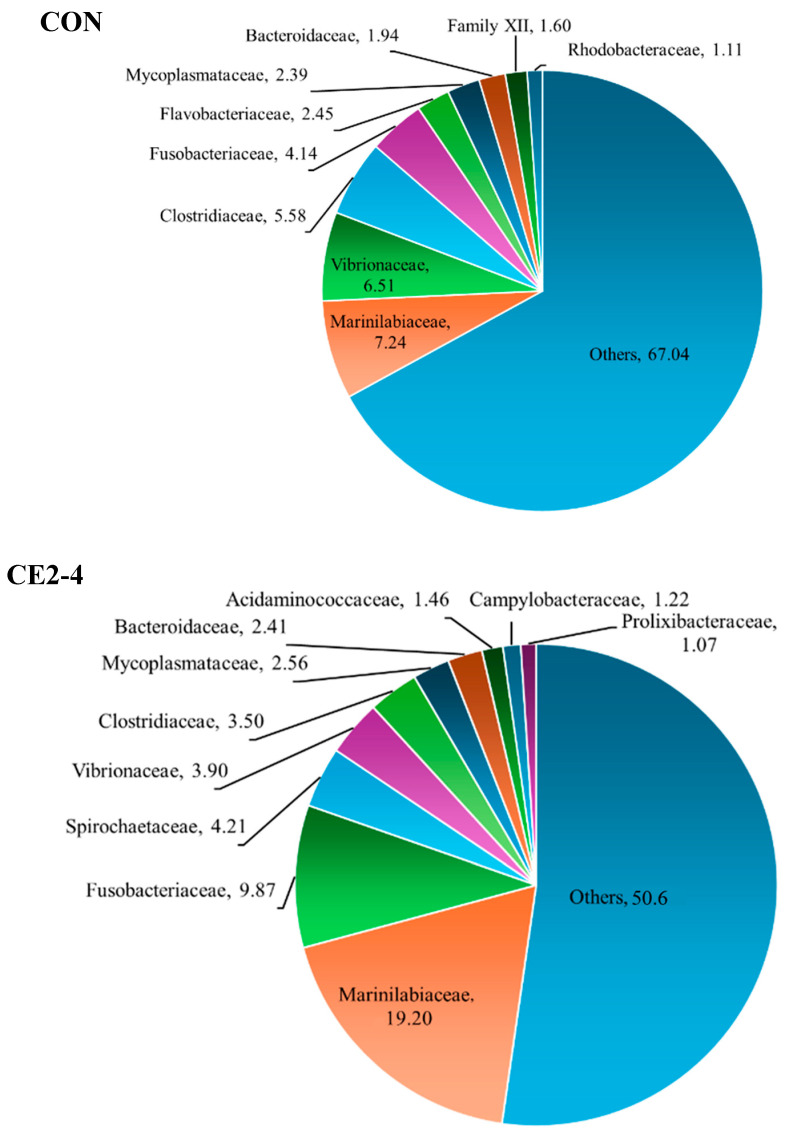
The family-based bacterial diversity of Asian green mussels (AGMs) depurated with water (CON) and 2% chitooligosaccharide-epigallocatechin gallate conjugate (CEC) on days 2 and 4, respectively. CON: mussels depurated using water after 2 days of storage. CE2-4: Asian green mussels depurated using a 2% CEC solution after 4 days of storage.

**Figure 7 foods-13-03104-f007:**
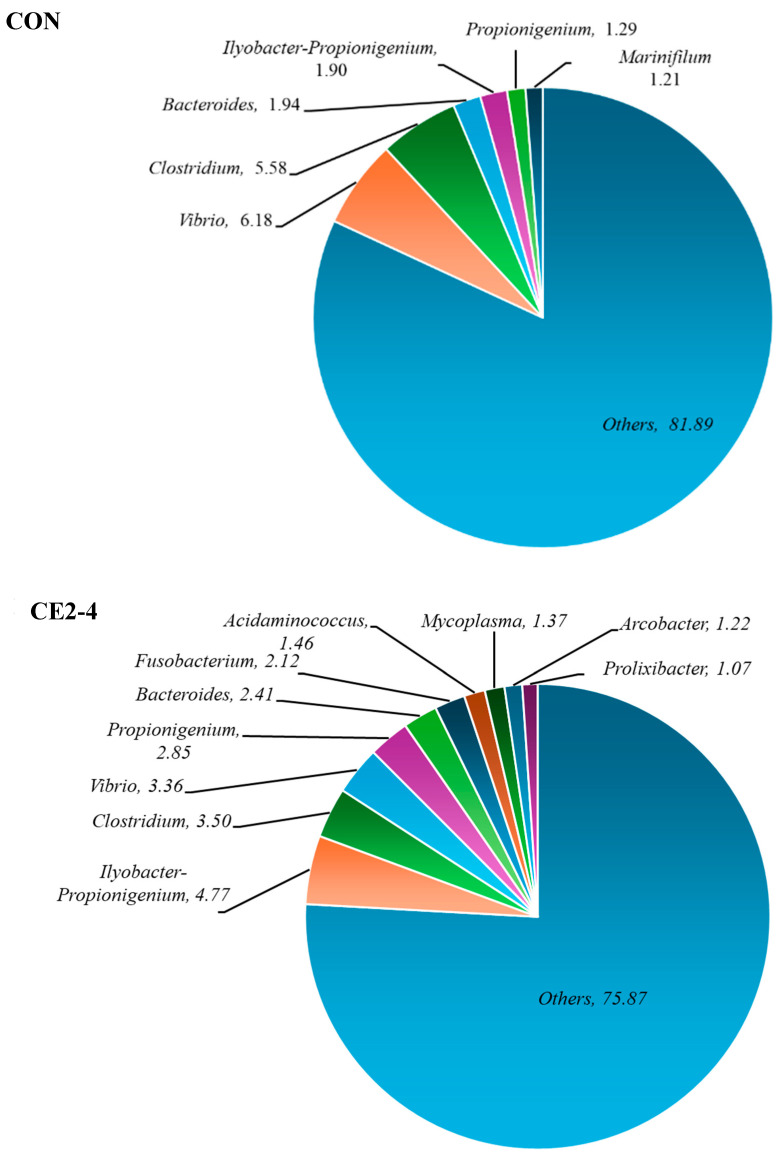
The genera-based bacterial diversity of Asian green mussels (AGMs) depurated with water (CON) and 2% chitooligosaccharide-epigallocatechin gallate conjugate (CEC) on days 2 and 4, respectively. CON: mussels depurated using water after 2 days of storage. CE2-4: Asian green mussels depurated using a 2% CEC solution after 4 days of storage.

**Figure 8 foods-13-03104-f008:**
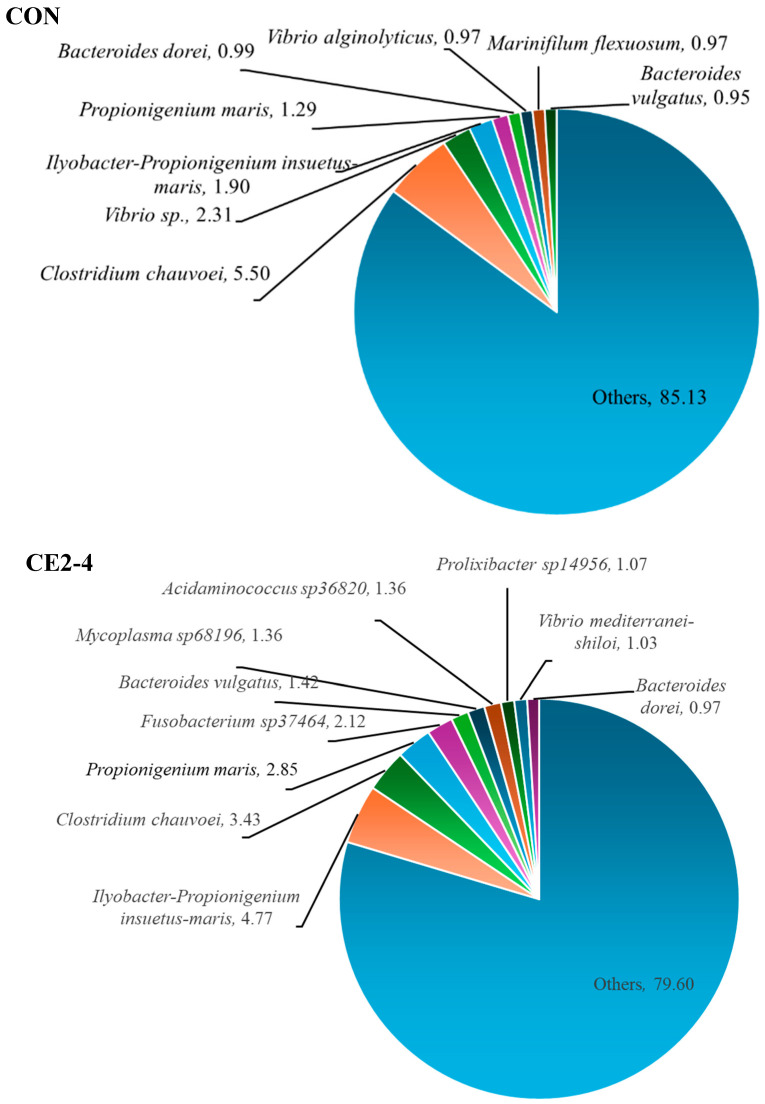
The species-based bacterial diversity of Asian green mussels (AGMs) depurated with water (CON) and 2% chitooligosaccharide-epigallocatechin gallate conjugate (CEC) on days 2 and 4, respectively. CON: mussels depurated using water after 2 days of storage. CE2-4: Asian green mussels depurated using a 2% CEC solution after 4 days of storage.

**Figure 9 foods-13-03104-f009:**
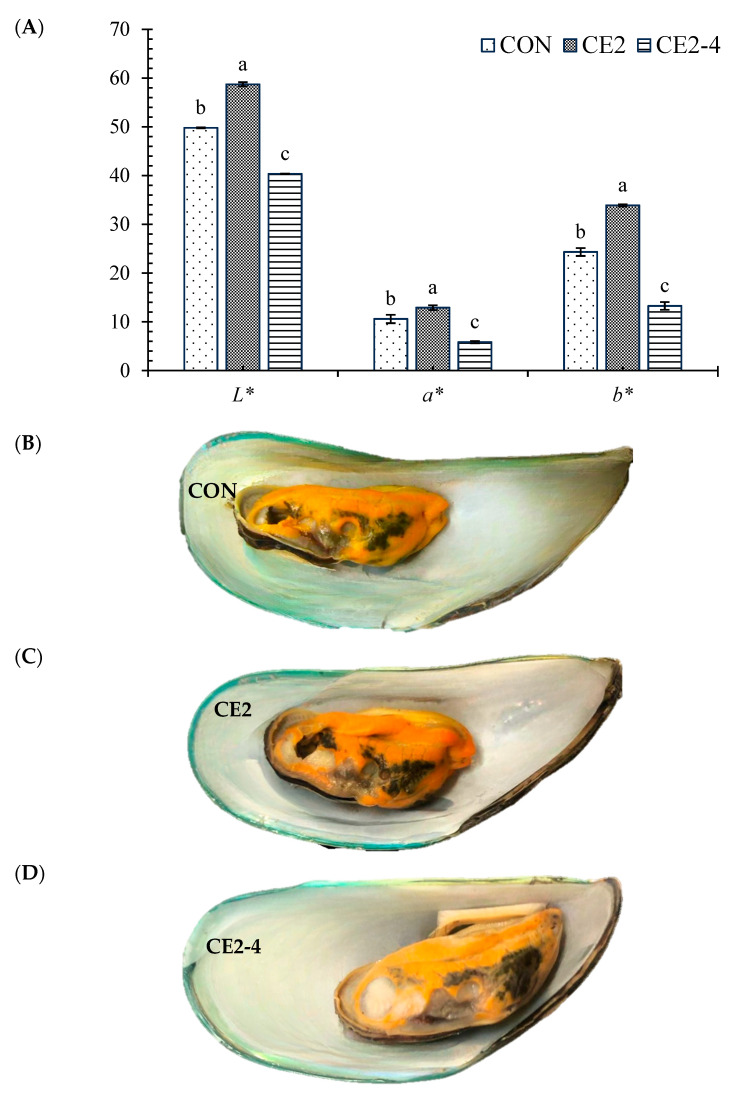
Color (**A**) and appearance (**B**–**D**) of steam-cooked Asian green mussels (AGMs) depurated with water (CON) and 2% chitooligosaccharide-epigallocatechin gallate conjugate (CEC) on days 0 and 4. Bars represent the standard deviation (*n* = 3). Lowercase letters on the bars of different samples within the same parameters indicate a significant difference (*p* < 0.05). CON: mussels depurated using water from day 0; CE2: mussels depurated using 2% chitooligosaccharide-epigallocatechin gallate conjugate (CEC) solution from day 0. CE2-4: Asian green mussels depurated using 2% CEC solution after 4 days of storage.

**Table 1 foods-13-03104-t001:** Fatty acid composition of Asian green mussels (AGMs) treated without and with COS-EGCG conjugate (CEC) during storage at 4 °C from day 0 and 4.

Fatty Acids (mg/g Lipids)	Day 0		Day 4
	CON	CE2	CE2–4
C14:0 (Myristic Acid)	43.62 ± 1.01 ^b^	50.55 ± 0.98 ^a^	38.27 ± 2.69 ^c^
C14:1 (Myristoleic Acid)	6.61 ± 0.66 ^a^	5.63 ± 0.48 ^a^	5.86 ± 0.66 ^a^
C15:0 (Pentadecanoic Acid)	8.66 ± 0.66 ^a^	8.35 ± 1.35 ^a^	9.35 ± 1.21 ^a^
C15:1 (cis-10-Pentadecanoic Acid)	27.01 ± 3.56 ^a^	17.63 ± 1.75 ^c^	19.87 ± 1.38 ^b^
C16:0 (Palmitic)	371.92 ± 5.61 ^a^	366.51 ± 11.63 ^a^	346.62 ± 9.06 ^b^
C16:1 (Palmitoleic Acid)	61.48 ± 3.04 ^b^	67.36 ± 4.10 ^a^	42.34 ± 3.84 ^c^
C17:0 (Heptadecanoic Acid)	40.27 ± 1.96 ^a^	41.84 ± 3.02 ^a^	23.41 ± 0.91 ^b^
C17:1 (cis-10-Heptadecanoic Acid)	14.03 ± 0.75 ^a^	14.74 ± 1.38 ^a^	13.02 ± 1.39 ^a^
C18:0 (Stearic Acid)	101.67 ± 1.86 ^a^	103.64 ± 3.26 ^a^	83.93 ± 4.41 ^b^
C18:1 (Oleic Acid)	56.72 ± 3.71 ^a^	59.69 ± 3.35 ^a^	46.00 ± 4.13 ^b^
C18:2 (Linoleic Acid)	6.69 ± 0.52 ^a^	6.99 ± 0.65 ^a^	6.45 ± 0.48 ^a^
C18:3 (gamma-Linolenic Acid)	52.29 ± 2.71 ^b^	57.51 ± 3.83 ^a^	42.79 ± 0.50 ^c^
C20:1 (cis-11-Eicosenoic Acid)	13.86 ± 0.69 ^a^	14.30 ± 1.27 ^a^	11.41 ± 0.92 ^b^
C20:0 (Docosanoic Acid)	9.79 ± 0.83 ^b^	10.70 ± 0.29 ^a^	8.99 ± 0.07 ^c^
C20:3 (cis-11,14,17-Eicosatrienoic Acid)	5.25 ± 0.40 ^a^	5.30 ± 0.08 ^a^	5.09 ± 0.29 ^a^
C20:4 (cis-5,8,11,14-Eicosatetraenoic Acid)	87.46 ± 2.37 ^b^	94.05 ± 4.96 ^a^	67.80 ± 5.11 ^c^
C23:0 (Tricosanoic Acid)	8.22 ± 0.47 ^b^	9.36 ± 0.72 ^a^	6.48 ± 0.50 ^c^
C20:5 (cis-5,8,11,14,17-Eicosapentaenoic Acid)	19.79 ± 0.46 ^a^	18.27 ± 1.95 ^a^	11.96 ± 0.74 ^b^
C22:6 (cis-4,7,10,13,16,19-Docosahexaenoic Acid)	10.48 ± 0.46 ^a^	9.79 ± 0.79 ^a^	7.99 ± 0.61 ^b^
C24:0 (Lignoceric Acid)	9.63 ± 0.67 ^a^	9.06 ± 0.10 ^a^	8.99 ± 0.05 ^b^
Saturated fatty acids	584.18 ± 12.43 ^a^	600.49 ± 21.39 ^a^	526.08 ± 18.94 ^b^
Monounsaturated fatty acids	179.74 ± 12.42 ^a^	179.38 ± 19.36 ^a^	138.52 ± 17.34 ^b^
Polyunsaturated fatty acids	181.99 ± 6.94 ^a^	191.94 ± 11.14 ^b^	142.10 ± 7.66 ^c^

The results are presented as means ± SD (*n* = 3). Different lowercase letters in the same row denote significant differences (*p* < 0.05). CON: mussels depurated using water; CE2 and CE2-4: mussels depurated using 2% chitooligosaccharide-epigallocatechin gallate conjugate (CEC) solution during storage at day 0 and 4, respectively.

**Table 2 foods-13-03104-t002:** Cooking loss, texture properties, and sensory analysis of cooked Asian green mussels (AGMs) treated without and with COS-EGCG conjugate (CEC) on days 0 and 4 when stored at 4 °C.

		Day 0		Day 4
		CON	CE2	CE2-4
Textural properties	Cooking Loss (%)	42.73 ± 1.57 ^a^	43.71 ± 2.06 ^a^	39.76 ± 1.77 ^b^
	Shear force (g/mm)	71.38 ± 1.23 ^a^	72.32 ± 2.36 ^a^	68.32 ± 2.02 ^b^
	Firmness (g)	80.32 ± 2.00 ^a^	79.36 ± 1.98 ^a^	76.02 ± 1.02 ^b^
	Toughness (g)	3801.34 ± 12.02 ^a^	3796.00 ± 11.78 ^a^	3502.32 ± 20.32 ^b^
Sensory analysis	Appearance	7.27 ± 0.46 ^a^	7.07 ± 0.96 ^a^	6.80 ± 0.77 ^b^
	Color	7.27 ± 0.88 ^a^	7.27 ± 1.10 ^a^	6.87 ± 1.06 ^b^
	Smell	6.6 ± 0.91 ^a^	6.07 ± 0.59 ^a^	6.47 ± 0.74 ^a^
	Texture	7.07 ± 0.96 ^a^	7.27 ± 0.88 ^a^	7.47 ± 0.83 ^a^
	Taste	6.93 ± 0.80 ^a^	7.13 ± 0.83 ^a^	7.07 ± 0.88 ^a^
	Overall	7.13 ± 0.99 ^a^	7.2 ± 0.94 ^a^	7.27 ± 0.88 ^a^

The results are presented as means ± SD (*n* = 3). Different lowercase letters in the same row denote significant differences (*p* < 0.05). CON: mussels depurated using water; CE2: mussels depurated using 2% chitooligosaccharide-epigallocatechin gallate conjugate (CEC) solution. CE2-4: Asian green mussels depurated using 2% CEC solution.

## Data Availability

The original contributions presented in the study are included in the article and Appendix A, further inquiries can be directed to the corresponding author.

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
