# Peer review of "Depuration of Asian Green Mussels Using Chitooligosaccharide-Epigallocatechin Gallate Conjugate: Shelf-Life Extension, Microbial Diversity, and Quality Changes during Refrigerated Storage"

_foods, 2024, doi:10.3390/foods13193104_

Round 1

Reviewer 1 Report

Comments and Suggestions for Authors

The manuscript described the effect of chitooligosaccharide-EGCG conjugate (CEC) at different concentrations (0, 1, and 2%; w/v) and depuration times (DT; 1, 3, and 6 h) on the shelf-life of Asian green mussels (AGM) during refrigerated storage at 4 ºC for 6 days. Generally, this manuscript was well written and logically conceived, and the results looks reasonable. The following points need to be addressed:

1. The significance analysis result of the main data (including data in Fig.3C, Fig.3D, Fig.3E and Fig.3F) should be provided to support the conclusions better.

2. Figure 6F should be given as clear picture.

3. Please, check the format of tables and figures according to journal requirements. And, the abbreviations used in the tables and figures should be explained in the footnote of tables and/or figure captions to make them standalone.

4. The Discussion is too simple, which is just like a study report. Hence, the comprehensive discussion, analysis and comparison should be made with other relevant studies.

Author Response

The manuscript described the effect of chitooligosaccharide-EGCG conjugate (CEC) at different concentrations (0, 1, and 2%; w/v) and depuration times (DT; 1, 3, and 6 h) on the shelf-life of Asian green mussels (AGM) during refrigerated storage at 4 ºC for 6 days. Generally, this manuscript was well written and logically conceived, and the results looks reasonable. The following points need to be addressed:

*****Thank you so much for your invaluable time spent on our manuscript for its better improvement. All the queries have been responded to and highlighted in yellow.

  1. The significance analysis result of the main data (including data in Fig.3C, Fig.3D, Fig.3E and Fig.3F) should be provided to support the conclusions better.

*****Thank you. The whole data has been compared using statistical software, wherever needed the results has been discussed based on the statistical analysis.
***** For Figure 3C to F, there was no difference in CFU was noticed as all of are bellow the 2 log CFU/g sample as mentioned in line 337. In such cases, it is reasonable to conclude that further statistical analysis may not be necessary because the low microbial counts do not warrant further examination of significant differences. This low level of growth likely supports the conclusion directly without additional statistical testing.

  1. Figure 6F should be given as a clear picture.

*****Thank you. The Figures has been updated for better clarity. Please see Figure captions from 6-8 and line 464.

  1. Please, check the format of tables and figures according to journal requirements. And, the abbreviations used in the tables and figures should be explained in the footnote of tables and/or figure captions to make them standalone.

*****Thank you. Authors have used the template from the journal itself, where all the format has been followed. For further confirmation, authors have cross-checked the format and necessary amendments have been made.

*****All the abbreviations have been already explained in Figure 1, 3 and 6. However, for better clarity, each figure caption has been explained with all abbreviations.

  1. The Discussion is too simple, which is just like a study report. Hence, a comprehensive discussion, analysis and comparison should be made with other relevant studies.

*****All the results and their discussion had been already made. However, considering the reviewers’ comments, some discussion has been extended as highlighted in yellow. Thank you.

Reviewer 2 Report

Comments and Suggestions for Authors

This manuscript investigated the effects of chitooligosaccharide and epigallocatechin gallate conjugate treatment on microbial and quality of Asian green mussels during storage at 4 ºC. The study identified the optimal treatment conditions. The manuscript is well-written, comprehensive, with accurate analysis, and appropriately presented graphs, leading to correct conclusions. However, there are minor issues that need improvement:

1.    Add a reference in line 55.

2.    The introduction needs improvement; more studies on the application of COS and EGCG or current studies on AGM depuration treatments should be added.

3.    The NGS method must be described in detail.

4.    In Figure 1B, 3-CON should not be AB, and 6-CE2 should not be B.

5.    In Figure 3A, 2-CON should be B. In Figure 3B, 0-CE1 should not be ac.

6.    The asterisks (****) in Figure 3 annotations are not explained. In the annotations for all figures, different letters indicate significant differences.

7.    It is recommended to include correlation analysis results of the indicators.

Author Response

This manuscript investigated the effects of chitooligosaccharide and epigallocatechin gallate conjugate treatment on microbial and quality of Asian green mussels during storage at 4 ºC. The study identified the optimal treatment conditions. The manuscript is well-written, comprehensive, with accurate analysis, and appropriately presented graphs, leading to correct conclusions. However, there are minor issues that need improvement:

*****Thank you so much for your invaluable time spent on our manuscript for its better improvement. All the queries have been responded to and highlighted in green.

  1. Add a reference in line 63.

*****Thank you, the reference has been added to the suggested sentence, please see line 63.

  1. The introduction needs improvement; more studies on the application of COS and EGCG or current studies on AGM depuration treatments should be added.

*****Thank you for the suggestions. The introduction has been updated as suggested. Please see line 52-57 and 67-73.

  1. The NGS method must be described in detail.

*****Thank you for the suggestions. The method has been explained. Please see section 2.6.

  1. In Figure 1B, 3-CON should not be AB, and 6-CE2 should not be B.

*****Sorry for the mistake. The Figure has been corrected.

  1. In Figure 3A, 2-CON should be B. In Figure 3B, 0-CE1 should not be ac.

*****Sorry for the mistake. The Figure has been corrected.

  1. The asterisks (****) in Figure 3 annotations are not explained. In the annotations for all figures, different letters indicate significant differences.

*****Thank you for the comments. The asterisks indicate the number of samples analyzed during each day. The annotations of different letters have been updated in all figure captions. Please the respective Figure captions.

  1. It is recommended to include correlation analysis results of the indicators.

*****Thank you for the comments, wherever it is possible authors had already conducted (line 402, 439, 476, 669) correlation between different indicators. Please see line However, considering the reviewers’ suggestions, text has been updated.

Reviewer 3 Report

Comments and Suggestions for Authors

This work entitled "Depuration of Asian green mussel using chitooligosaccharide-epigallocatechin gallate conjugate: shelf-life extension, micro-bial diversity, and quality changes during refrigerated storage” by Buatong et al., constitutes effects of chitooligosaccharide-epigallocatechin gallate conjugate on the shelf-life of treated and untreated Asian green mussel. Overall the manuscript is interesting and might attract interest of the people working in the similar realm.

Some specific points for the manuscript is highlighted below:

Absence of Context: The entirety of the data fails to mention why this study is important. It is crucial to provide a comparative data with specific standard antimicrobial agents. Simply providing the effects of chitooligosaccharide-epigallocatechin gallate conjugate without its comparison with standard antimicrobials does not give importance of the tested products.  

- Lack of Details: The study does not provide sufficient methodological details, e.g. methods used for the GC-FID analysis is not provided in the experimental section.

-Authors should include the Chromatograms of the GC-FID analysis of fatty acids.

- Authors should include the retention time for the determined fatty acids along with kovats index of the identified compounds.

-Conclusion is unnecessary long. This section should summarize the deliverables or the implications of the results. It is essential to provide a synthesis of the key perspectives and their significance.

Comments on the Quality of English Language

Minor editing of English language required.

Author Response

This work entitled "Depuration of Asian green mussel using chitooligosaccharide-epigallocatechin gallate conjugate: shelf-life extension, micro-bial diversity, and quality changes during refrigerated storage” by Buatong et al., constitutes effects of chitooligosaccharide-epigallocatechin gallate conjugate on the shelf-life of treated and untreated Asian green mussel. Overall, the manuscript is interesting and might attract interest of the people working in the similar realm.

*****Thank you so much for your invaluable time spent on our manuscript for its better improvement. All the queries have been responded to and highlighted in pink.

Some specific points for the manuscript are highlighted below:

Absence of Context: The entirety of the data fails to mention why this study is important. It is crucial to provide a comparative data with specific standard antimicrobial agents. Simply providing the effects of chitooligosaccharide-epigallocatechin gallate conjugate without its comparison with standard antimicrobials does not give importance of the tested products.

***** The authors acknowledge the reviewer’s suggestion to include a comparison with standard antimicrobial agents. However, the focus of this study aligns with the growing demand for green technology and natural bioactive agents, as many scientists are increasingly exploring alternatives to synthetic antimicrobials. Among these alternatives, chitooligosaccharide (COS) has emerged as a promising candidate due to its antioxidant and antimicrobial properties.

******In this study, we have specifically investigated the COS-EGCG conjugate, derived from shrimp waste, in its application to extend the shelf life of Asian green mussels (AGM). While the antimicrobial efficacy of standard agents is well-documented, our previous research has already highlighted the antioxidant and antimicrobial activities of COS and its conjugates (refer to references). Therefore, our primary goal was to evaluate the effectiveness of COS-EGCG in a real-world application, which we have successfully demonstrated through the shelf-life extension of AGM in comparison to untreated controls. To justify this focus, the text has been updated to reflect the rationale behind excluding direct comparisons with standard antimicrobial agents (see line 75-81).

******Thus, considering our objective of exploring alternative antimicrobial agents, the authors did not conduct a comparison with known synthetic antimicrobial agents. However, your comments are well considered, and we will implement in our currently going on study where non-thermal processing has been applied. Thank you for the insightful comment.

- Mittal, A.; Singh, A.; Zhang, B.; Visessanguan, W.; Benjakul, S. Chitooligosaccharide Conjugates Prepared Using Several Phenolic Compounds via Ascorbic Acid/H2O2 Free Radical Grafting: Characteristics, Antioxidant, Antidiabetic, and Antimicrobial Activities. Foods 2022, 11, 920. https://doi.org/10.3390/foods11070920

- Lack of Details: The study does not provide sufficient methodological details, e.g. methods used for the GC-FID analysis is not provided in the experimental section.

******Thank you for the comments. The methods were shortened to avoid plagiarism, which is really hard to remove from the methodology. However, considering the reviewer comments, authors have provided all methodology. Please see the methodology section 2.5 highlighted in pink.

-Authors should include the Chromatograms of the GC-FID analysis of fatty acids.

******Thank you for the comments. For ease of understanding and to present the data in a more concise and accessible format, we have provided the fatty acid composition as mg/g of lipids instead of chromatograms. This approach allows for a direct comparison of fatty acid content across samples and simplifies the interpretation of the data.

*****Considering the reviewers suggestions chromatograms were also provided in the supplementary file.

- Authors should include the retention time for the determined fatty acids along with kovats index of the identified compounds.

*****There might be slight misunderstanding for the fatty acid analysis as the full method was not provided, which has been updated. Please see section 2.5. The Kovats Index is typically calculated using a homologous series of n-alkanes as standards to express retention times relative to the number of carbon atoms in the compound. However, in our study, we used the Supelco 37 Component Fatty Acid Methyl Ester (FAME) Mix, which contains C4-C22 fatty acid methyl esters in the concentration range of 50 to 400 µg/mL, as our standard. The use of the FAME mix was selected as it is directly relevant to our target analytes, allowing for precise identification and quantification of fatty acids.

Separation and retention times of the fatty acid methyl esters in the FAME mix, spanning from C4:0 (butyric acid) to C22:6 (docosahexaenoic acid), were obtained under the conditions detailed in the Agilent method (SI-02178), which provides a robust separation protocol specifically for FAMEs. Using this standard enabled us to match the retention times of our analytes directly with the corresponding FAME components, facilitating accurate analysis within the context of fatty acid profiling.

Although the Kovats Index is a useful tool for standardizing retention times across different conditions, its calculation requires the use of n-alkane standards, which were not part of our experimental setup. In this study, the use of the FAME standard is more appropriate and widely accepted for fatty acid analysis, ensuring high precision and relevance to the analytes under investigation.

-Conclusion is unnecessary long. This section should summarize the deliverables or the implications of the results. It is essential to provide a synthesis of the key perspectives and their significance.

****The conclusion has been amended to reflect key results. Please see the respective section.

Round 2

Reviewer 1 Report

Comments and Suggestions for Authors

1.In order to better support the conclusion, Figure 3C、3D、3E and 3F need to provide specific data, and it is suggested to change the form of tables.

2.The “p” and/or “p” value should be italics, please check the full text and be consistent.

Author Response

*****Authors express their sincere thanks to the reviewer for their critical analysis of our manuscript. All the queries have been responded and highlighted in yellow.

  1. In order to better support the conclusion, Figure 3C、3D、3E and 3F need to provide specific data, and it is suggested to change the form of tables.

*****Thank you for the comments, Authors did not observe the bacterial colonies on the respective plates of Clostridium perfringens count (CC), lactic acid bacterial count (LAB), and Enterobacteriaceae count (EC), therefore it is not possible to provide data. However, to avoid confusion all those figures has been removed, except HSPBC, whose data has been provided and discussion has been extended in line 286, 299-300, and 336-341.

*****For table format changes has been updated as per the guidelines: Title heading in the table is bolded, and font size change to 10 for the main text. Whereas footnote font size to 9. Please see the highlighted tables. Thank you for the comments

  1. The “p” and/or “p” value should be italics, please check the full text and be consistent.

****Thank you for the suggestion, all the “p” font type in the whole text has been changed as suggested by the reviewer.